# Promoting Health and Behavior Change through Evidence-Based Landscape Interventions in Rural Communities: A Pilot Protocol

**DOI:** 10.3390/ijerph191912833

**Published:** 2022-10-07

**Authors:** Shan Jiang, Udday Datta, Christine Jones

**Affiliations:** 1GBBN Architects, Pittsburgh, PA 15206, USA; 2School of Design and Community Development, West Virginia University, Morgantown, WV 26506, USA; 3Community Care of West Virginia at Big Otter (Big Otter Clinic), Ivydale, WV 25113, USA

**Keywords:** evidence-based design, therapeutic landscapes, physical activity, rural community, health promotion

## Abstract

Rural communities in the United States have many public health issues, including a high prevalence of physical inactivity, obesity, and higher risks for major non-communicable diseases. A lack of safe and convenient places to exercise could intensify healthy lifestyle disparities. Individually adapted physical activity prescriptions at the primary level of healthcare could play a role in behavior change for rural residents. Healthcare professionals and designers created the rural wellness hub concept, which integrates walking trails and therapeutic landscape features on the clinic site, to support patient physician-prescribed activities and treatments. This research protocol reports the design and implementation of the rural wellness hub at a clinic in Clay County, West Virginia. Following a participatory, evidence-based landscape intervention (EBLI) protocol, 58 user representatives (patient = 49; clinic employee = 9) participated in the four-phase protocol: (1) pre-design survey, (2) design and development, (3) post-design interview, and (4) post-occupancy evaluation. Survey and interview data from all phases were collected and analyzed. The preliminary results indicate that the redesigned clinic campus could promote several health programs among local communities, with the benefits of walking trails, in particular, highlighted. The rigorous EBLI protocol could serve as a template for rural communities that seek to develop similar healthcare intervention programs.

## 1. Introduction

Sedentary lifestyles and physical inactivity have been associated with many chronic diseases and a significant portion of death in the United States [1]. Physical inactivity prevalence among rural populations is much higher than their non-rural counterparts [2]. Public health professionals have been advocating behavior interventions that focus on physical activity promotion, in which the core intervention programs should match the unique attributes and preferences of the target population [3]. Individually adapted physical activity prescriptions at the primary level of healthcare have emerged as a new method to engage people and deliver positive outcomes; however, physicians are facing multiple obstacles, including a lack of access to resources and facilities, and inadequate coordination of the health intervention programs [4]. Therapeutic landscapes in clinical environments as a health intervention have started to gain attention in contemporary research. This article reports an evidence-based landscape intervention (EBLI) protocol that designs and develops a rural wellness hub in support of physical activity prescriptions and healthy lifestyle promotion.

### 1.1. Physical Inactivity Prevalence in Rural Communities

Physical inactivity for adults is defined as not participating in any regular leisure-time physical activities such as running, walking for exercise, or gardening [5]. According to a series of studies conducted by the Centers for Disease Control and Prevention (CDC) [6], seven states in the U.S. (i.e., West Virginia, Oklahoma, Louisiana, Alabama, Kentucky, Arkansas, and Mississippi) and Puerto Rico have a physical inactivity prevalence of 30% or more. A lack of safe and convenient places to exercise in those areas could intensify disparities in leading a healthy lifestyle [6]. The World Health Organization (WHO) [7] has warned that physically inactive lifestyles are a leading cause of major non-communicable diseases and can seriously impact people’s health status. Strong evidence has shown significant correlations between physical inactivity and the increased risk of obesity, coronary heart disease, diabetes, cancer, and shorter life expectancy [8].

West Virginia is categorized as one of the most rural states in the U.S., with more than 50% of its population living in rural communities [9]. The state has been ranked as the 11th state with high physical inactivity prevalence—more than 28.5% of West Virginian adults did not routinely participate in leisure-time physical activity or exercise [10]. Despite the highly ranked forest coverage and abundant natural resources, the West Virginian rural population—particularly seniors, the disabled, and the medically underserved—have limited daily access to public open spaces and physical activity opportunities [11,12]. According to a cross-sectional study on more than 400 participants who participated in primary care, the most common barrier to physical activity was lack of resources (80.5%), which was particularly significant among people of lower income socioeconomic status [13]. Gilbert et al. [14] made similar conclusions that the physical activity participation rate among rural populations is significantly lower than urban and suburban counterparts, and environmental barriers were significant obstacles to physical activities, including the lack of desired amenities and design characteristics, location, accessibility, and safety.

In the context of emerging needs for safe, accessible, and convenient sites to host various health intervention programs, the rural wellness hub concept was coined collectively by healthcare professionals and designers, and first piloted in southern West Virginia. The hub integrates walking trails and therapeutic landscape features to support physician-prescribed physical activities and treatments on the clinic site. The design and development of the hub followed a rigorous protocol, namely that of evidence-based landscape intervention (EBLI), which involves stakeholder participation and research components to ensure the hub’s usability and wellness outcomes.

### 1.2. Behavior Change through Community Building

Several behavioral change models contribute to the theoretical framework for the EBLI protocol. The Health Belief Model (HBM) was originally developed by a group of social psychologists in the U.S. Public Health Service who aimed to detangle a series of applied research problems that had emerged since the 1950s, and has been expanded and updated to serve as a dominant model in health behavior change [15,16,17]. Among the six key constructs of HBM, the “perceived barriers” were negative aspects of a particular health action that may act as obstacles for individuals undertaking recommended behaviors [18]. The “cues to action” suggest that cues such as bodily events and media publicity could help trigger the actions to perform the recommended behavior [18]. The theory of planned behavior (TPB) focuses on explaining individual motivation as a decisive factor in performing a specific behavior—a person’s strong beliefs that positive outcomes are associated with the behavior (i.e., behavioral belief) determines the individual’s willingness to perform the behavior [19]. A person’s perceived control over behavioral performance, including the perceived ease or challenge, is expected to have a direct effect on executing the behavior [20].

Rural communities hold unique cultures that vary in their approach to physical activities and health behaviors [14,21]. “Important determinants of health-related behavior are embedded in relationships that tie individuals to organizations, neighborhoods, families, and friends in their community” [22]. Therefore, building rural community organizations with accessible resources and collective goals could be effective models for facilitating health behavior change among rural populations [23]. Recreational and fitness programming in indoor facilities have been suggested to promote physical activity and healthy lifestyles for rural communities [24]. Rural communities in regions that have heavy industries such as coal mining and steel industry face serious challenges regarding environmental health [25]. Green public open spaces in rural communities in West Virginia will promote residents’ exposure to healthy environments and lead to positive health outcomes [26]. Considering the numerous benefits of green exercise (i.e., participating in physical activities in a natural setting) on people’s physical and mental health [27], implementing therapeutic landscapes and outdoor features in rural wellness centers as interventions to promote physical activities and wellbeing deserves further exploration.

### 1.3. Evidence-Based Design Approach

Evidence-based design (EBD) is a process of planning, designing, and constructing healthcare environments that makes decisions according to credible research to achieve the best possible outcomes [28]. The EBD has proven its effectiveness in promoting patient and user experiences and medical outcomes in the healthcare context, which has become a leading trend and effective tool in the healthcare design realm [29]. Eight key steps in EBD span the entire cycle of the healthcare project: (1) defining the EBD goals and objectives, (2) identifying sources of information, (3) critical interpretation of relevant research evidence, (4) the generation of innovative EBD concepts, (5) hypothesis development and testing, (6) collecting baseline performance measures, (7) monitoring design implementation and construction, and finally (8) conducting post-occupancy evaluations [28]. Along the different phases of an EBD project, sources of reliable information should cover relevant research evidence in the knowledge field as well as key stakeholders’ and end-users’ opinions to ensure that the information can be appropriately interpreted in line with the cultural norms of local communities [30]. The essence of EBD is user-centered, which maximizes the usability of the design for optimal outcomes. In situations where gaps exist between research evidence and practices, ad hoc studies with research questions and hypotheses arising from the project could be strategically embedded in the EBD process. Using community-based, non-empirical samples to address those questions and hypotheses could aid decision-making by providing loops of feedback quickly in a well-situated context [31,32].

### 1.4. Research Objectives

This study aimed at the development and implementation of an EBLI protocol that converts a traditional rural clinic to a wellness hub that facilitates several physical activity intervention programs directly on the clinic site. Primary research objectives addressed in the study include: (1) to explore suitable physical activities and design features that fulfill users’ needs on the clinic site. (2) to implement the EBLI protocol to support the rural wellness hub concept on a pilot clinic site.

## 2. Materials and Methods

### 2.1. The Pilot Site and Context

The Community Care of West Virginia (CCWV) Big Otter Clinic in Clay County is the pilot site for this study. The CCWV organization is a federally qualifying health clinic network that serves low-income and underinsured patients in West Virginia. The CCWV Big Otter Clinic (henceforth referred to as the clinic) sought to renovate its outdoor environments by introducing a series of landscape features to support treatments and physical activities and to host educational programs for local communities in the county. The need to renew the existing clinic grounds arose in response to two emergent healthcare approaches: (1) the clinic physicians intended to prescribe exercises and walking time for their primary care patients and coordinate physical activity programs directly on the clinic site to improve participation [3]; and (2) the clinic therapists intended to offer walk-and-talk therapy sessions to facilitate patients’ psychological processing [33]. Therefore, the rural wellness hub concept was introduced to the clinic site not just to deliver sickness treatment but to support disease prevention and wellness care. The wellness hub will provide safe and accessible opportunities for people to walk, exercise, and meditate in nature; the hub includes paths for walking and exercise, seating areas with different levels of privacy, and gathering spaces for community events and educational programs. The clinic physicians and staff members can also use the outdoor space to take breaks from work-related stress [34,35]. The ultimate project goal is to transition a traditional clinic site to a wellness center to serve as a node in an extensive network of trails and parks in the region.

### 2.2. The Four-Phased EBLI Protocol

The project follows a mixed-method, four-phase EBLI protocol that integrates key components of evidence-based design and community-based participatory design, as discussed, including (1) pre-design survey, (2) site design and development, (3) post-design interview, and (4) post-occupancy evaluation (POE). The pre-design survey was designed to explore user’s preferred outdoor activities, desired landscape features, behavior patterns, and overall usage situations of the clinic site. Patients and clinic staff members were randomly recruited to form a representative sample of users. Two versions of the survey questionnaires were administrated, including a hard copy survey for the patients and an online version for the employees. Each version of the survey had some tailored questions for the user group; for instance, the patient participants were asked questions such as “how often do you visit the clinic?” and “how do you spend your waiting time before seeing the doctor?” Comparatively, employee participants were asked questions such as “how much break time do you have per day?” and “how do you spend your break time?” Both surveys asked the same questions about the participant’s preferred physical and leisure activities, and their preferred landscape design features using 5-point Likert scales.

The site design and development phase were conducted by the project team consisting of faculty and graduate students in landscape architecture from West Virginia University. The design team conducted a thorough site analysis and a literature review of relevant research evidence, then utilized the pre-design survey to inform the initial site design. After completing the first round of site design, a post-design interview (Phase 3) was conducted to collect user representatives’ feedback on the initial design ideas. The initial site design plan, together with a series of reference images depicting different design elements, were presented to the interviewees. Participants’ preference for different programming elements and design styles was recorded, transcribed, and analyzed by the project team. The design suggestions were looped back to reshape the site design and development and finalize the Phase 2 efforts (Figure 1). There could be multiple rounds of post-design interviews with different emphases depending on the scope of the project and the level of details. Frequent feedback and quick assessments during design implementation and site construction are also desirable from the knowledge-sharing perspective [36]. The pre-design surveys and post-design interview questions can be accessed as supplementary materials through the open access data deposit [37].

Due to funding constraints, the clinic prioritized the implementation of walking trails and phased development of other features over future years. As reported in February 2022, the outside border trail and benches along the trail were completed and some minimal landscaping along the site border was under construction. The next round of funding will be used to install additional seating, signage, and to develop the meditation garden near the behavioral health office. The trails have been used by patients and physicians and the clinic doctor has provided some written feedback as preliminary POE data (Phase 4). The project team plans to conduct a full POE after the site construction is completed and has been experienced by the users.

### 2.3. Methods and Participants

A total of 58 user representatives (patient = 49; clinic employee = 9) participated in the four-phased EBLI protocol, including 48 patients and 6 clinic employees during Phase 1 pre-design survey, 1 patient representative and 1 employee representative during Phase 3 post-design interview, and 2 employees who provided preliminary POE feedback during Phase 4. Participants in the project were protected by the Office of Research Integrity and Compliance at West Virginia University.

#### 2.3.1. Pre-Design Survey

*Patient Survey*. The project was conducted during the COVID-19 pandemic with restricted in-person visiting policies. Therefore, the clinic doctor helped coordinate the pre-design survey. Considering the potential of the digital divide among rural and senior populations, a paper-pencil survey questionnaire was administrated to patient participants onsite [38]. Patients who visited the clinic between September 2020 and February 2021 were recruited by the front desk receptionist and asked to fill out the questionnaire onsite if the patient agreed to participate in the survey. The survey period followed the project funding period and the site design and construction timeline. The number of survey invitations and the response rate was not traceable.

A convenience sample of 48 adult patients participated in the pre-design survey and the demographic information of the sample is provided in Table 1. Participants in the sample belong to diverse age groups but were skewed to the seniors, with 4.2% from 18–24 years old, 6.3% from 25–34 years old, 18.8% from 35–44 years old, 8.3% from 45–54 years old, 25% from 55–64 years old, and 37.5% who were 65 years of age or older. The sample was dominated by white (97.9%) and female (83.3%) participants. The majority, 72.9% of the participants, were regular patients of the clinic, and 20.8% were occasional clinic visitors when they need to see a doctor. The sample’s age distribution was representative as the clinic doctor described the clinic as serving mostly senior patients. Looking at the broad picture, the Big Otter clinic serves Clay County in West Virginia. According to the census data estimation as of July 2021, the county’s population estimates was 7892, with 22.3% senior residents (65 years and above) and 49.1% female residents; about 97.6% of the county’s residents were white alone, and 23.3% of the county residents were in poverty [39].

*Employee Survey*. The web-version of the pre-design survey was distributed to all clinic employees through the internal email system, including three behavioral health specialists and nine general clinic staff. Six clinic employees participated in the pre-design survey and the demographic information of the sample is shown in Table 1. The employee survey response rate was about 66.7%. Employee participants in the sample belong to three age groups, with 16.7% from 25–34 years old, 50% from 35–44 years old, and 33.3% from 45–54 years old. All employee participants were white females (Table 1).

#### 2.3.2. Post-Design Interview and Content Analysis

All participants in the survey phase were asked if they agreed to be contacted for Phase 3 of the study. Among those who indicated willingness to participate, a total of two user representatives (1 patient and 1 nurse) showed up for post-design interviews (Phase 3) through one-on-one Zoom meetings on 26 March 2021. Transcribed interviews were analyzed using content analysis techniques aided by QDA Miner Lite software [40]. Transcribed interviews were coded in four main categories: (1) Design Favored, (2) Design Unfavored, (3) Functionality, and (4) Interaction. Under each category, multiple themes emerged. These themes were used to analyze each interview to understand the participants’ responses to the proposed designs.

#### 2.3.3. Preliminary POE Feedback

The site construction was not fully completed when the protocol was documented, and only one doctor provided preliminary POE feedback through conventional email communications. A diagnostic POE study will be conducted after the site is fully constructed [41].

### 2.4. Statistical Analyses

Descriptive statistics (mean [M], standard deviation [SD]) and one-way between-group analysis of variance (ANOVA) with Tukey’s HSD Test for multiple comparisons were conducted to compare the differences between age groups regarding the preferred physical activities and site design features. All statistical analyses were conducted using the IBM SPSS (Version 28.0) software (IBM, Armonk, NY, USA) [42].

## 3. Results

### 3.1. Pre-Design Survey

#### 3.1.1. Patient Survey

The pre-design survey results served as guidelines to indicate design decisions. Patient participants were asked to rate their level of preference for a series of recreational activities on 5-point Likert scales (1 = “Dislike very much”, 5 = “Like very much”). One response was excluded due to missing values and eventually, 47 responses were analyzed. Referring to Sullivan and Artino [43], “parametric tests are sufficiently robust to yield largely unbiased answers that are acceptably close to ‘the truth’ when analyzing Likert scale responses,” therefore, the mean preference scores of the survey items are reported below.

The top five preferred activities were reading (M = 4.49, SD = 0.93), roaming in the woods (M = 4.4, SD = 1.1), gardening (M = 4.38, SD = 1.05), hiking (M = 4.26, SD = 1.21), and music/instruments (M = 4.06, SD = 1.21), and the bottom five preferred activities were climbing (M = 2.6, SD = 1.33), jogging/running (M = 2.79, SD = 1.37), yoga (M = 2.98, SD = 1.28), art (M = 3.21, SD = 1.35), and cycling (M = 3.28, SD = 1.31). Because age was a major variable in the study sample, the participants’ preferred recreational activities were analyzed and compared across different age groups. Figure 2 demonstrates that as the age increases, participants’ general interest in intensive physical activities decreased. A one-way between-group ANOVA was conducted to explore the impact of age on preference scores for recreational activities. The results indicated statistically significant differences for jogging/running, climbing, and reading among different age groups, with jogging *F*(6, 40) = 6.28, *p* < 0.001; climbing *F*(6, 40) = 4.49, *p* = 0.001; and reading *F*(6, 40) = 3.54, *p* = 0.007. Tukey’s HSD Test for multiple comparisons found that the mean preference scores for participants in age groups above 65 years old were significantly lower than participants under 45 years old for jogging and climbing, and the mean preference scores for reading among participants older than 65 years were significantly higher than participants in younger age groups (Table 2).

Participants were asked to rate their level of preference for five landscape programming elements for the site on 5-point Likert scales (1 = “Dislike very much”, 5 = “Like very much”). The average scores for the programming elements, ranked from the most to least preferred were: trails (M = 4.91, SD = 0.35), picnic areas (M = 4.68, SD = 0.73), community garden (M = 4.49, SD = 0.95), gathering space (M = 4.06, SD = 1.19), water feature (M = 3.87, SD = 1.21), and playground (M = 3.72, SD = 1.31). Figure 3 demonstrates that trails were ranked as the most preferred programming element by all age group participants, and the playground was ranked the least preferred by participants under 25 or older than 55 years old. A one-way between-group ANOVA indicated a statistically significant difference for playground among participants from different age groups, *F*(6, 40) = 2.76, *p* = 0.024, and post hoc tests did not identify statistical significance between different age groups.

#### 3.1.2. Employee Survey

Participants were asked to rate their level of preference for a series of recreational activities on 5-point Likert scales (1 = “Dislike very much”, 5 = “Like very much”). Due to the small sample size, only descriptive statistics were conducted on their preferred recreational activities and landscape programming. The top five preferred activities were cycling (M = 4.83, SD = 0.41), music or instruments (M = 4.83, SD = 0.41), roaming in the woods (M = 4.67, SD = 0.52), hiking (M = 4.17, SD = 0.75), and yoga (M = 4, SD = 1.55), and the bottom five referred activities were art (M = 3.5, SD = 0.84), gardening (M = 3.5, SD = 1.05), jogging or running (M = 3.5, SD = 1.23), reading (M = 3.5, SD = 1.05), and climbing (M = 3.67, SD = 1.03). Employee participants were also asked to rate their level of preference for five landscape programming elements. The average scores for the programming elements, ranked from the most to least preference were: playground (M = 5, SD = 0), community garden (M = 4.67, SD = 0.82), trails (M = 4.5, SD = 0.55), gathering space (M = 4.5, SD = 0.55), and water feature (M = 3.33, SD = 1.51).

#### 3.1.3. Summary and Additional Feedback

To summarize the pre-design survey results from all user groups, the site design should emphasize trails that support walking and roaming in the woods, places to sit or read, gardening in community gardens, and multifunctional spaces for music or similar community events. Additionally, the patient participants reported their feedback about the quality of the clinic’s built environment through open-ended questions. Nineteen participants contributed their suggestions regarding the features they wish to improve for the clinic. The top-ranked items were all relevant to the outside environments, including landscapes/landscaping (6 times/31.6%), adding trails on the clinic ground (3 times/15.8%), and more outside lighting (2 times/10.5%). Other minor suggestions included interior paint and flooring, parking lot, waiting room layout, and modernizing the overall architectural style (1 time/5.3% for each). Employees reported having 15–30 min of break time during a typical workday in the clinic with moderate stress level. Their average satisfaction score with the overall physical environment of the clinic was neutral (M = 3.4, SD = 0.89), and their desired landscape programming elements included: places to sit outside for lunch, trails to walk before or after work, a more naturalistic design, and more lighting outside.

### 3.2. Post-Design Interview

The patient representative had an overall positive response to the initial site design proposal. Handicap accessibility and ease for wheelchair users were the primary concerns. The overall mood of the place was very important. The participant preferred designs that helped create a warm, welcoming, joyous environment. In terms of functionality, the participant welcomed the inclusion of outdoor lighting, wayfinding signage, an outdoor bathroom, and a healing garden. A colorful planting palette with a naturalistic style was preferred over designs with geometrical shapes and hardscape elements. The participant thought a dense, shaded, and secluded type of trail looked lonely and preferred an open parkland style to encourage socialization. Water features were also preferred because of their calming effect. Design elements such as firepits were not considered fitting as they can be hazardous for small children accompanying the elderly. The participant welcomed the idea of a hammock garden and thought it could be a good place to read. Design elements such as a greenhouse and a yoga or meditation area were welcomed. Noise from the adjacent road was identified as a major problem, and a noise buffer was suggested.

The employee representative also had a positive response to the initial preliminary site design plan. The safety of the elderly occupants and ease of maintenance were priorities for selecting detailed design style and elements. The naturalistic landscaping was preferred over high-maintenance geometric styles. The participant welcomed the inclusion of water features due to their soothing impact on the users and the seasonal wet conditions of the site were identified as a potential design opportunity to create an immersive landscape. A mixture of socializing spaces and spaces that allow users to relax and contemplate might be ideal for this project. Low maintenance traditional planter boxes at a raised height were favored. Naturalistic landscaping and a vivid color palette were preferred over manicured lawns. The public park style outdoor furniture was preferred over homey style patio-type furniture. Design elements such as fire pits and hammock gardens were considered unfitting for an older generation target population. The presence of a power line going across the proposed helipad location was identified as a significant limitation.

### 3.3. Preliminary Post Occupancy Evaluation (POE)

Two clinic doctors reported their observations of the site usage as preliminary POE data through email communications from February to September 2022. The behavior health doctor reported that they have started using the therapeutic landscape features and walking trails for the walk-and-talk therapy: “Our counselor sees a lot of benefit with the school-age children when she takes them outside. We also have had positive feedback from the adult patients as well.”

One family medicine doctor provided some feedback about the trail usage situation by different patient groups (Figure 4A–C):

“The park, even in the beginning stages, is being used already. Our behavioral health team find the walking appt options very useful for ADHD children. They feel they can get 10–15 min of solid time with them in a much more relaxed atmosphere. We hosted a fundraiser for breast cancer on the site in October. The staff is using it for walking at lunch…I have begun a walking club on Tuesday afternoon with patients, although this is weather dependent in the winter. We have had some feedback already that everyone wants the outside trail to be paved so they can take wheelchairs.”—Big Otter Clinic Doctor Feedback

## 4. Discussion

The rural population in the U.S. is facing increased rates of physical inactivity, obesity, and the associated health risks for major non-communicable diseases [44]. Infrastructure interventions, such as trails and parks where outdoor exercises and healthful events are hosted, could play an important role in promoting physical activity and raising health awareness among local communities [45]. For rural communities that are isolated or lack safe and convenient places to exercise and host those healthful events, it becomes imperative to establish a network of rural wellness centers that consolidate the resources and extend public health support. This study documented the process that transformed a traditional rural clinic to a wellness hub from ideation to design and implementation following the EBLI protocol. The protocol emphasized design decision-making and iteration informed by research evidence and direct input by the community users. Pre-design survey, post-design interview, and preliminary post-occupancy evaluation was conducted to collect representative users’ opinions on different landscape programming and design features.

When offering outdoor recreational opportunities, sites should be chosen carefully considering users’ preferences and attitudes. The pre-design survey revealed that participants’ preferred recreational activities and landscape design features vary significantly by age group, which corresponds with previous research that found a reduction in physical activity intensity and functional fitness was associated with the aging process [46]. Generally, major site users favored activities and design features such as trails and roaming in the woods, places to sit or read, and gardening in community gardens.

The post-design interview results indicated that different user groups have different design concerns: the clinic employee prioritized the daily maintenance of the site, and the patient was concerned with wheelchair users’ ability to access and use the design features. Different opinions about water features emerged from the pre-design survey and the post-design interview. Because the sample size was small for the post-design interview, users’ attitudes about water features on a clinic site requires further exploration in future studies.

Preliminary POE feedback demonstrated additional functions of the site beyond the initial intent. The walking trail could offer a relaxed atmosphere in facilitating treatments during walk-and-talk therapy sessions. Existing studies have indicated that ongoing professional guidance in a face-to-face format may improve the effectiveness of physical activity programs [47,48]. The individually adapted physical activity prescriptions guided by the clinic doctor, such as the patient walking club, could be an effective behavioral intervention.

The small sample size using a convenience sampling technique was a limitation of the study. There have been similar studies that used smaller sample sizes and gained reliable research insights [49,50]. Considering the EBLI protocol was still in the pilot testing stage, the sample of patients and employees recruited directly from the clinic site provided meaningful insight to guide the design and implementation of the wellness hub [51]. The protocol will need to be validated using a larger sample size in various rural locations.

The wellness hub concept is not new to rural communities; some hubs were piloted on school sites [52], and others were centered on parks and recreation opportunities [53]. This protocol intended to fill the gap between therapeutic landscapes, physical activity intervention programs, and a physician-led wellness hub on a rural clinic site, with an emphasis on the evidence-based design and evaluation of the site features from user-centered perspectives. The implementation of the EBLI protocol was completely a bottom-up process that involved multiple stakeholders, including the core design-research team, community participants, public university extension programs, and numerous nonprofit funding agencies. Gathering community input and feedback during all phases of the project was important to the successful construction of the physical environment of the hub. Recognizing that each user group has its own unique culture and needs sheds light on the equitable access and inclusiveness of the rural wellness hub.

## 5. Conclusions

The EBLI protocol emphasized participatory design, community building, and cultural uniqueness when implementing behavioral intervention programs for rural communities. It suggested walking trails and other age-appropriate programs that converted a traditional clinic site into a wellness hub in a rural county of West Virginia. The protocol could serve as a template for the design and development of a rural wellness hub that integrates therapeutic landscape features to support physical activities through doctor prescriptions and community-level health intervention programs. The effectiveness of various intervention programs hosted by the wellness hub, such as the patient walking club and walk-and-talk therapy programs, need systematic measurements and documentation in the next step of the study.

## Figures and Tables

**Figure 1 ijerph-19-12833-f001:**
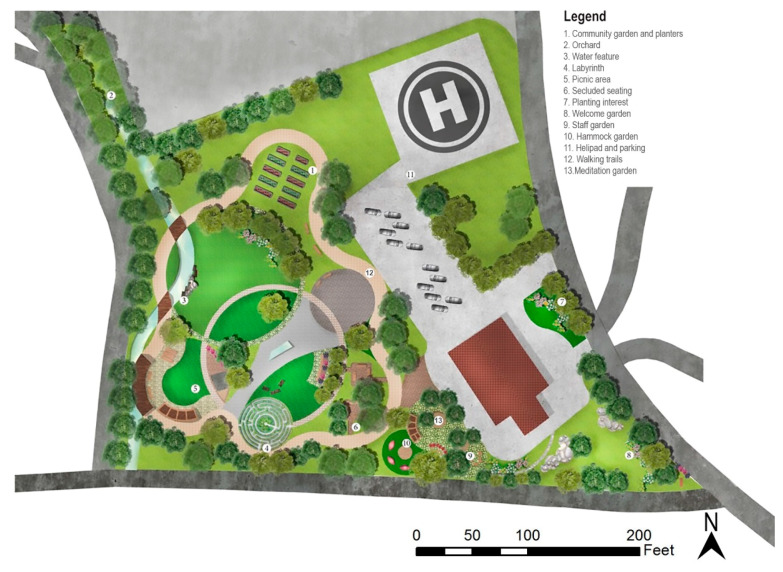
Finalized site plan for the clinic after Phase 2 and 3 following the EBLI protocol.

**Figure 2 ijerph-19-12833-f002:**
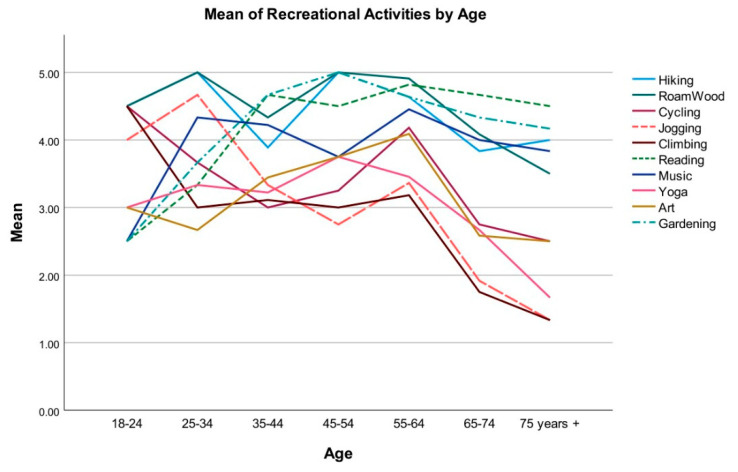
Preference of recreational activities by age.

**Figure 4 ijerph-19-12833-f004:**
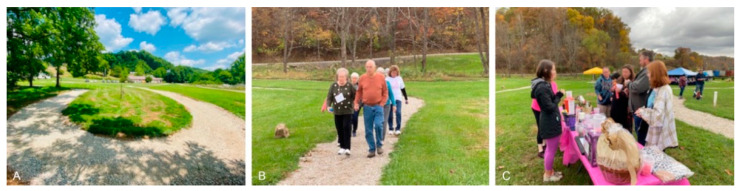
(**A**) The first loop of trails installed on the clinic site; (**B**) trails supporting the patient walking club organized by the clinic doctor; and (**C**) the fundraising event for breast cancer hosted on the site in October 2021.

**Figure 3 ijerph-19-12833-f003:**
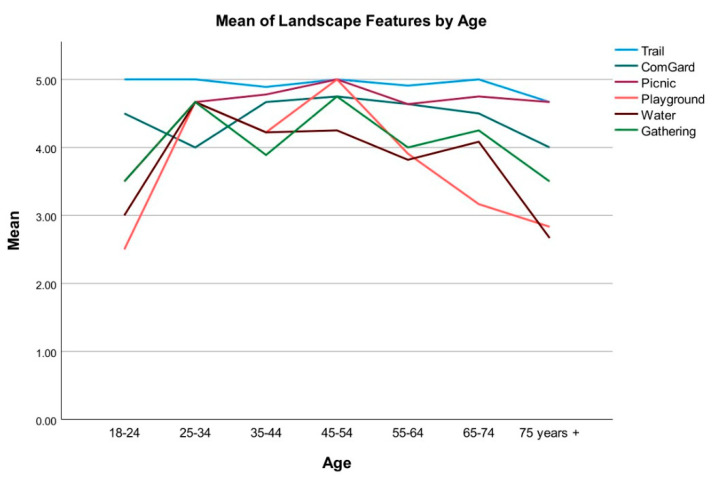
Preference of landscape features by age.

**Table 1 ijerph-19-12833-t001:** Survey participants demographic information.

Demographic Information		Patient Survey Participants	
		Count	Valid Percent
Age	18–24 years old	2	4.2
	25–34 years old	3	6.3
	35–44 years old	9	18.8
	45–54 years old	4	8.3
	55–64 years old	12	25
	65–74 years old	12	25
	75+ years old	6	12.5
Gender	Male	8	16.7
	Female	40	83.3
Ethnicity	White	47	97.9
	Native American/ American Indian	1	2.1
Clinic Visit	Regular patient	35	72.9
	Occasionally as needed	10	20.8
	Emergency	2	4.2
	Missing data	1	2.1
**Demographic Information**		**Staff Survey Participants**	
		**Count**	**Valid Percent**
Age	25–34 years old	1	16.7
	35–44 years old	3	50
	45–54 years old	2	33.3
Gender	Female	6	100
Ethnicity	White	6	100

**Table 2 ijerph-19-12833-t002:** Tukey’s HSD Test for Multiple Comparisons by Recreational Activities. This table reports only the statistically significant results because of the great number of variables and categories.

Dependent Variable	(I) Age	(J) Age	Difference (I–J)	Std. Error	Sig.	95% Confidence Interval	
						Lower Bound	Upper Bound
Jogging or Running	18–24	75+	2.67	0.86	0.05	0.00	5.33
	25–34	65–74	2.75	0.68	0.00	0.64	4.86
		75+	3.33	0.74	0.00	1.03	5.64
	35–44	75+	2.00	0.55	0.01	0.28	3.72
	55–64	65–74	1.45	0.44	0.03	0.09	2.81
		65–74	1.45	0.44	0.03	0.09	2.81
		75+	2.03	0.53	0.01	0.37	3.69
Climbing	18–24	65–74	2.75	0.84	0.03	0.14	5.36
		75+	3.17	0.90	0.02	0.37	5.96
	55–64	65–74	1.43	0.46	0.05	0.00	2.86
		75+	1.85	0.56	0.03	0.11	3.58
Reading	18–24	35–44	−2.17	0.63	0.02	−4.12	−0.21
		55–64	−2.32	0.62	0.01	−4.24	−0.40
		65–74	−2.17	0.62	0.02	−4.08	−0.26

## Data Availability

The study’s supplementary materials are available online with open access: https://github.com/UddayDatta/Pilot_Protocol_supplementary_materials/tree/main.

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
