# Peer review of "Promoting Health and Behavior Change through Evidence-Based Landscape Interventions in Rural Communities: A Pilot Protocol"

_ijerph, 2022, doi:10.3390/ijerph191912833_

Round 1

Reviewer 1 Report

The subject is relevant. The implementation of healing gardens is a new vast and mostly undiscovered field for contemporary research. The study documented the process that transformed a  rural clinic to a wellness center following the EBLI protocol. The paper well-written and interesting.

Just a few minor remarks:

Please consider attaching the pre-design survey questions (both hard copy and online versions), post-design interview questionnaire, and post-occupancy evaluation (POE) survey.

It would be beneficial to  readers, who want to replicate this study in different locations.

Line 158, please specify the XXX University

Please consider adding a line or two, maybe a paragraph, to discuss the preferences of users demonstrated by the pre-design survey. What are the conclusions for healing garden design?

Author Response

Please see my reply to the review comments report attached. Thanks!

Reviewer 2 Report

Dear Authors,

Thank you for the opportunity to review your interesting paper.

I would recommend to improve:

(i) Abstract: please, make it more specific. 

(ii) Introduction: please, specify the objectives and the research question of your study.

(iii) Materials and Methods: the inclusion criteria for the respondents are missing. Please, extend the statistics analysis description. 

(iv) Results: actually, the description of the methods should be moved to the Materials and Methods section.

(v) Discussion: this section is very poor.

(vi) Conclusions: please, update and make relevant to the objectives of your study. 

Author Response

(The authors gave the same response as above.)

Reviewer 3 Report

AUTHORS AFFILIATION

- It is not correct: Affiliation #1 should be the same for authors Jiang S and Datta U (superscript 1 for both authors); so, for Jones C supercript should be #2.

INTRODUCTION

- A paragraph which integrates the three fields (1.1, 1.2, 1.3) is needed.

- Study justification, hypothesis and goals are missed, or they are not properly described.

MATERIALS AND METHODS

- After seeing the site plan in Figure 1, specific facilities related to exercise or wellness are not observed.

- Characteristics description of project team members is missed; at least, they should be mentioned in an acknowledgment section at the end of the manuscript. Why is the University of the project team members anonymized along the text? Any of the team members are the co-authors of this manuscript?

- Figure 1 should be located after being mentioned in the text (after line 170).

- Research methodology is missing, or is not mentioned in the manuscript: Ethics Committee authorization, informed consents signed, participants recruitment procedure, inclusion criteria, exclusion criteria, abandonment criteria, number of subjects/group required (why n=48 in patients group, and n=6 in clinic staff group?), survey design and items, interview design and questions, statistical analysis...

- Methods are not described.

RESULTS

- The random recruitment of sample does not allow to reflect the real sample distribution (age group, sex) at the clinic, both in patients and staff. 83,3% patients are women? And 100% employees are women?

- Statistically significant differences are not highlighted, neither in tables nor in figures.

- Post-design interview: Considering 54 subjects participated in the pre-design survey, why only 2 subjects in the post-design interview? This is not explained or justified. Why not more subjects, or everyone who had participated previously in the pre-design survey?

- Line 291: "2 user representatives (1 patient and 1 nurse) [...]". Which were the criteria to be selected as representative?

DISCUSSION

- Lines 371-373: "Generally, trails and roaming in the woods, places to sit or read, and gardening in community gardens were favored activities and design features by major site users". These facilities are not directly related to physical inactivity or obesity, which are mentioned in the Introduction and Discussion".

- Lines 377-378: "Different opinions about water features emerged from the pre-design survey and post-design interview [...]". This is understandable, because the pre survey was answered by 56 subjetcs, and the post interview by 2 subjects.

CONCLUSION

- There is a not an indicators table which allows to demonstrate the goals achievement rate; thus, the intervention program effectiveness can not be assessed.

Author Response

(The authors gave the same response as above.)

Reviewer 4 Report

Thank you for giving me this opportunity to read the manuscript entitled "Promoting Health and Behavior Change through Evidence-Based Landscape Interventions in Rural Communities: A Pilot Protocol". The topic of this manuscript is interesting and would be a good contribution to this field. I think it could be considered for publication in International Journal of Environmental Research and Public Health once the following issues are addressed.

  1. I suggest that the authors add some review of theories or research on environmental health to the Introduction, as the manuscript is focusing on Heath, Behavior and Landscape.

  1. The research questions is also suggested to be raised in the last section in Introduction.

  1. Lines 112 -113: Some newly published papers are encouraged to be cited here, for example a paper titled “Dynamic assessments of population exposure to urban greenspace using multi-source big data.” Can be used to support the point statement here.

  1. The scale and compass need to be enlarged so that readers can clearly read text information.

  1. Limitations should be added and well discussed as a sub-section in the Discussion section.

  1. Some grammatical errors exist in the manuscript. Therefore, a critical review of the manuscript language will improve readability.

Author Response

(The authors gave the same response as above.)

Reviewer 5 Report

The manuscript aims to describe an evidence-based landscape intervention (EBLI) protocol and its implementation in a rural healthcare clinic.  The authors suggest the EBLI can be a template for other localities, particularly rural areas, with limited availability of places for exercise.  While the manuscript clearly describes the implementation process of the protocol, some details are missing.

Major Concerns:

Lines 185-186:   Justification should be provided for the use of only 1 patient representative and 1 staff representative for Phase 3.

Line 203-204:  Was the Phase 1 clinic sample’s demographic distribution reflective of the clinic patient population beyond the age distribution?  What about the distribution of the staff demographics?  What is the total N of the clinic staff?

Lines 210-213:  Address the normality (or lack thereof) of the data.

Lines 222-224:  Move the statistical methods to the Methods section of the paper.

In general, limitations encountered during the implementation of the protocol should be noted as well as any lessons learned for future implementations.

While the focus of the manuscript is on the implementation of the protocol, the final outcome measure and the time frame for measurement should be noted.

Minor Concerns:

Table 1:  I would suggest removing the “Cumulative Percent” column and the “Total” rows since neither adds information.  Adding a column of comparable statistics to the Patient part of Table 1 from the US Census for the county would be beneficial.

Figures 2 & 3:  Consider a different type of graph or color/line scheme to describe the data.  The colors of the lines are very difficult to distinguish.  Also remove “Multiple Line Mean” from the titles.

Author Response

(The authors gave the same response as above.)

Round 2

Reviewer 2 Report

Dear Authors,

Thank you for your efforts to improve the text.

Some minor comments:

(i) lines 135-137: One research answers one research question. These research questions sound moslly like objectives

(ii) you do not need to underline the words in the text (lines 214, 234)

(iii) line 265: please, specify the producer of the software (country and company)

(iv) lines 475-480: Conclusions are poor. Please, improve it and connect with your aims and research question(s).

Author Response

Please see our response in the attached document.

Reviewer 3 Report

A complete review and re-writing from authors is evident. These changes and implementations along the text mean a significant improvement with regard to scientific rigour.

For this new evaluation, "Objectives and research question" inclusion, "Material and Methods" and "Conclusions" sections re-writing have been basic.

Just one thing to correct: number of participants is different in some parts.

- Abstract: 58 (49 patients + 9 employees).

- Methods and Participants (2.3 section): 57 (49 patients + 8 employees).

- Table 1: 54 (48 patients + 6 employees).

An homogenization in this number along the full text, in every section, is needed.

Author Response

(The authors gave the same response as above.)

Reviewer 5 Report

By the authors addressing the limitations and providing additional information in the text, the manuscript grew stronger and more comprehensive.   The increase in size of Figure 1 made it readable.  

The only unaddressed comment is in regarding the population of the county or of the medical practice beyond age.  To gain understanding of the interpretability of the survey, one needs to understand how reflective the convenience survey is of the population.  All that is provided is an opinion of the physician on age.  Also, I would still suggest removing the "Total" rows from Table 1.  They are redundant and not necessary.

Author Response

(The authors gave the same response as above.)
